# Optimization of individualized faricimab dosing for patients with diabetic macular edema: Protocol for the SWAN open-label, single-arm clinical trial

**Takao Hirano**[1]*, **Toshinori Murata**[1], **Shintaro Nakao**[2], **Masahiko Shimura**[3], **Miho Nozaki**[4], **Kiyoshi Suzuma**[5], **Taiji Nagaoka**[6], **Masahiko Sugimoto**[7], **Yoshihiro Takamura**[8], **Tomoaki Murakami**[9], **Keisuke Iwasaki**[10], **Jun Tsujimura**[10], **Shigeo Yoshida**[11]

1 Department of Ophthalmology, Shinshu University School of Medicine, Nagano, Japan, 2 Department of Ophthalmology, Juntendo University School of Medicine, Tokyo, Japan, 3 Department of Ophthalmology, Tokyo Medical University Hachioji Medical Center, Tokyo, Japan, 4 Department of Ophthalmology, Laser Eye Center, Nagoya City University East Medical Center, Aichi, Japan, 5 Department of Ophthalmology, Kagawa University Faculty of Medicine, Kagawa, Japan, 6 Department of Ophthalmology, Asahikawa Medical University, Hokkaido, Japan, 7 Department of Ophthalmology, Yamagata University Faculty of Medicine, Yamagata, Japan, 8 Department of Ophthalmology, Faculty of Medical Sciences, University of Fukui, Fukui, Japan, 9 Department of Ophthalmology and Visual Sciences, Kyoto University Graduate School of Medicine, Kyoto, Japan, 10 Chugai Pharmaceutical Co., Ltd, Tokyo, Japan, 11 Department of Ophthalmology, Kurume University School of Medicine, Fukuoka, Japan

* takaoh@shinshu-u.ac.jp

## Abstract

### Purpose

In patients with diabetic macular edema (DME) from YOSEMITE/RHINE, dual angiopoietin-2/vascular endothelial growth factor-A (VEGF-A) inhibition with faricimab resulted in visual/anatomic improvements with extended dosing. The SWAN trial (jRCTs031230213) will assess the efficacy, durability, and safety of faricimab during the treatment maintenance phase in patients with DME using a treat-and-extend (T&E)-based regimen adapted to clinical practice and the characteristics of patients achieving extended dosing intervals.

### Methods

SWAN is a 2-year, open-label, single-arm, interventional, multicenter trial enrolling adults with center-involving DME. All patients will receive three initial faricimab 6.0 mg doses every 4 weeks (Q4W). From week 12 onwards, in patients without active DME, dosing intervals will be extended in 8-week increments up to Q24W. In contrast, patients with active DME (central subfield thickness [CST] >325 μm and intraretinal fluid [IRF] or subretinal fluid [SRF] resulting in vision loss/disease aggravation) will receive a dose within a day and the dosing interval will be shortened by 4 weeks to a minimum of Q8W relative to the previous dosing interval. Recruitment commenced in August 2023 across a planned 16 sites in Japan.

relevant data from this study will be made available upon study completion.

**Funding:** This study was supported by Chugai Pharmaceutical Co., Ltd. The funder participated in the study design, data collection and analysis, the decision to publish, and in the preparation of the manuscript.

**Competing interests:** Takao Hirano: Speaker Fees: Bayer, Canon, Chugai Pharmaceutical Co., Ltd., Kowa, Novartis Pharma KK, Santen Pharmaceutical Co., Senju, ZEISS Toshinori Murata: Speaker Fees: Bayer, Novartis, Santen, Zeiss Meditec: Consultant: Boehringer Ingelheim, Chugai Pharmaceutical Co., Ltd., Hoya, Kowa, Wakamoto Shintaro Nakao: Consultant: Alcon, Boehringer Ingelheim, Novartis, Riverfield; Speaker Fees: Bayer, Boehringer Ingelheim, Chugai Pharmaceutical Co., Ltd., Hoya, Kowa, Machida, Mitsubishi Tanabe, Novartis, Novo Nordisk, Otsuka, Santen, Senju, Wakamoto Masahiko Shimura: Consultant: Bayer, Boehringer Ingelheim, Chugai, HOYA, Nikki HD, Roche, Wakamoto; Lecture Fees: Bayer, Chugai, Kowa, Novartis, Otsuka, Senju Miho Nozaki: Speaker Fees: Bayer, Canon, Chugai Pharmaceutical Co., Ltd., Kowa, Nikon, Novartis Pharma KK, Santen, Senju, Sumitomo Pharma Co., Ltd., Topcon Medical, Wakamoto Kiyoshi Suzuma: Speaker Fees: AMO, Alcon, Bayer, Chugai Pharmaceutical Co., Ltd., HOYA, Kowa, Novartis, Senju Consultant: Senju, Chugai Pharmaceutical Co., Ltd., Boehringer Ingelheim Taiji Nagaoka: Consultant: Boehringer Ingelheim, Novartis, TES Holdings; Speaker Fees: Bayer, Chugai Pharmaceutical Co., Ltd., Hoya, Kowa, Mitsubishi Tanabe, Novartis, Santen, Senju, Wakamoto; Research Fees: Daicel, Kowa, LTT, Santen Masahiko Sugimoto: Research Fees: Alcon Japan, Bayer, Chugai Pharmaceutical Co., Ltd., Novartis; Speaker Fees: Alcon Japan, Bayer, Kowa, Novartis, Senju, Wakamoto Yoshihiro Takamura: Lecture Fees: Chugai Pharmaceutical Co., Ltd. Tomoaki Murakami: Speaker Fees: Bayer, Canon, Chugai Pharmaceutical Co., Ltd., Johnson & Johnson, Kowa, Novartis Pharma KK, Santen; Consultant: Boehringer Ingelheim Keisuke Iwasaki: Employee: Chugai Pharmaceutical Co., Ltd. Jun Tsujimura: Employee: Chugai Pharmaceutical Co., Ltd. Shigeo Yoshida: Consultant: Chugai Pharmaceutical Co., Ltd., Novartis; Speaker Fees: Bayer, Chugai Pharmaceutical Co., Ltd., Novartis, Senju; Research Fees: Kowa, Otsuka, Senju We confirm that the author competing interests do not alter our adherence to PLOS ONE policies on sharing data and materials.

## Results

The primary endpoint is change in best-corrected visual acuity (BCVA) from baseline at 1 year (averaged over weeks 52, 56, and 60). Key secondary endpoints include: change from baseline in BCVA, CST, and National Eye Institute Visual Function Questionnaire scores over time; proportion of patients with BCVA (decimal visual acuity) $\geq 0.5$, $\geq 0.7$, $\geq 1.0$, or $\leq 0.1$; proportion of patients with absence of DME, and IRF and/or SRF over time. Safety endpoints include incidence/severity of ocular/nonocular adverse events.

## Conclusions

The SWAN trial is expected to provide evidence to support individualized faricimab dosing regimens, with the potential to reduce the burden of frequent treatments on patients, caregivers, and healthcare systems.

## Introduction

Diabetic macular edema (DME) is a leading cause of vision impairment in people with diabetes [1]. Anti-vascular endothelial growth factor (VEGF) agents are widely used to treat DME; however, such treatment demands regular injection and monitoring visits, which may be a substantial burden for patients and their caregivers and affect treatment adherence [2, 3]. There is a need for more durable treatments that require fewer healthcare interactions without compromising effectiveness or patient safety [3, 4].

Faricimab is a humanized bispecific immunoglobulin G1 antibody that selectively binds to both VEGF-A and angiopoietin-2 (Ang-2) with high affinity. Faricimab is approved for the treatment of neovascular age-related macular degeneration, diabetic macular edema, and retinal vein occlusion in multiple countries worldwide, including Japan. The global, phase 3, randomized YOSEMITE (NCT03622580) and RHINE (NCT03622593) clinical trials (study design: see S1 Fig) of faricimab met their primary endpoints, with faricimab 6.0 mg dosed every 8 weeks (Q8W) and according to a personalized treat-and-extend (T&E) regimen demonstrating noninferior best-corrected visual acuity (BCVA) gains from baseline at 1 year (averaged over weeks 48, 52, and 56) compared with aflibercept 2.0 mg Q8W in patients with DME. Notably, over 70% of patients treated with faricimab T&E achieved a dosing interval of $\geq$ every 12 weeks (Q12W) by the end of the first year [5], while vision gains and anatomic improvements with faricimab after 1 year were maintained through 2 years [6]. Strong durability was also demonstrated at the end of the second year, with 78.1% of patients in the faricimab T&E arm on $\geq$Q12W dosing and every 16 weeks (Q16W) dosing achieved by 62.3% of patients [6]. Results from a subgroup analysis of patients in Japan enrolled in YOSEMITE through 1 year [7] and 2 years [8] were generally consistent with the global YOSEMITE and RHINE results. Furthermore, a subsequent YOSEMITE and RHINE *post hoc* analysis found that 56% of patients who achieved Q16W faricimab dosing met the criteria for potential extension to every 20 weeks (Q20W) dosing [9].

In clinical practice in Japan, patients often receive one initial anti-VEGF dose followed by as-needed treatment. Vision outcomes have been reported to be worse and treatment frequency lower in clinical practice compared with clinical trials, possibly because of less well-defined retreatment criteria and reactive regimens [10]. Findings from clinical trials clearly show that frequent treatment results in better vision outcomes; however, in clinical practice,

frequent treatment places a burden on patients, carers, and healthcare professionals, and disease activity assessment can be complicated. As such, there is a need for a treatment regimen applicable to routine clinical practice that improves vision outcomes, reduces treatment burden, and involves simplified disease activity assessment.

The SWAN trial will assess the efficacy, durability, and safety of faricimab during the treatment maintenance phase in patients with DME using a T&E dosing regimen adapted to clinical practice. The SWAN trial will also explore the characteristics of patients for whom dosing intervals can be extended in an effort to help optimize individual treatment in clinical practice and reduce treatment burden.

## Methods

### Trial overview

SWAN (jRCTs031230213) is a 2-year, open-label, single-arm, interventional, multicenter clinical trial. The trial will be carried out at a planned 16 sites in Japan, with the site of the primary sponsor located at Shinshu University Hospital. More sites may be added.

The trial will be conducted according to the principles set in the Declaration of Helsinki, and will also comply with local guidance, including the Clinical Trials Act, Ordinance for the Enforcement of Clinical Trials Act, and Ethical Guidelines for Medical and Health Research Involving Human Subjects. Written informed consent will be obtained before patients participate in this trial. The schedule of enrollment, interventions, and assessments is summarized in Fig 1.

### Participants and eligibility criteria

Eligible patients will be aged ≥18 years with a documented diagnosis of type 1 or 2 diabetes mellitus. One eye will be designated as the study eye. If both eyes are deemed to be eligible, the eye with the worse BCVA at screening will be selected as the study eye unless the investigator determines that the other eye is more appropriate for research treatment.

Key inclusion and exclusion criteria for the study eye are shown in Table 1. Ocular inclusion criteria include macular thickening secondary to DME involving the center of the fovea, with central subfield thickness (CST) ≥325 μm as measured on Spectralis spectral-domain optical coherence tomography (SD-OCT) (or ≥315 μm as measured on Cirrus or Topcon SD-OCT), BCVA of 0.0625 ~ 0.7 (decimal visual acuity), and sufficiently clear optic media and adequate pupillary dilatation to allow for good quality color fundus photography (CFP) and other retinal imaging at screening. Ocular exclusion criteria for the study eye include high-risk proliferative diabetic retinopathy, tractional retinal detachment, preretinal fibrosis, and vitreomacular traction syndrome or epiretinal membrane involving the fovea or disrupting the macular architecture. Patients with prior panretinal photocoagulation, macular laser, or cataract surgery within 3 months of day 1, intravitreal or periocular corticosteroid treatment within 6 months of day 1, prior intravitreal faricimab treatment, or prior anti-VEGF treatment at any time (for treatment-naïve eyes) or within 3 months (for previously treated eyes) of day 1 are also ineligible.

Previous intravitreal anti-VEGF treatment (>3 months from day 1) will be permitted for the study eye. Enrollment of study eyes with a history of intravitreal anti-VEGF treatment will be restricted to no more than 25% of the total study cohort. Full inclusion and exclusion criteria are shown in S1 Table (see Supporting information).

### Treatment protocol

Patients will receive three initial faricimab 6.0 mg doses Q4W, with dosing from week 12 onwards determined according to the presence/absence of active DME (Fig 2). Active DME is

| | Enrollment | Allocation | Post-allocation (week) | | | | | | | | | | | | | | | | | | End |
| --- | --- | --- | --- | --- | --- | --- | --- | --- | --- | --- | --- | --- | --- | --- | --- | --- | --- | --- | --- | --- | --- |
| TIMEPOINT | Screening Day −28 to Day 1 | Week 0 Day 1 | 4 | 8 | 12 | 16 | 20 | 24 | 32 | 40 | 48 | 52 | 56 | 60 | 64 | 72 | 80 | 88 | 96 | 104 | 112 |
| **ENROLLMENT:** | | | | | | | | | | | | | | | | | | | | | |
| Eligibility screen | x | x | | | | | | | | | | | | | | | | | | | |
| Informed consent | x | | | | | | | | | | | | | | | | | | | | |
| **Allocation** | | x | | | | | | | | | | | | | | | | | | | |
| **INTERVENTIONS:** | | | | | | | | | | | | | | | | | | | | | |
| **Faricimab** | | x | x | x | According to 8-week or 4-week extension regimen based on DME activity → | | | | | | | | | | | | | | | | |
| **ASSESSMENTS:** | | | | | | | | | | | | | | | | | | | | | |
| Visual acuity test | x | x | x | x | x | x | x | x | x[1] | x[1] | x[1] | x | x | x | x[1] | x[1] | x[1] | x[1] | x[1] | x[1] | x |
| Indirect ophthalmoscopy | x | x | x | x | x | x | x | x | x[1] | x[1] | x[1] | x | x | x | x[1] | x[1] | x[1] | x[1] | x[1] | x[1] | x |
| Color fundus photography | x | x | x | x | x | x | x | x | x[1] | x[1] | x[1] | x | x | x | x[1] | x[1] | x[1] | x[1] | x[1] | x[1] | x |
| FA | x | x* | | | x | | | | | | | x† | x† | x† | | | | | | | x |
| Indocyanine green angiography‡ | x | x | | | x | | | x | | | | x | x | x | | | | | | | x |
| SD-OCT | x | x | x | x | x | x | x | x | x[1] | x[1] | x[1] | x | x | x | x[1] | x[1] | x[1] | x[1] | x[1] | x[1] | x |
| OCT-angiography | | x | | | x | | | | | | | x† | x† | x† | | | | | | | x |
| IOP | x | x | x | x | x | x | x | x | x[1] | x[1] | x[1] | x | x | x | x[1] | x[1] | x[1] | x[1] | x[1] | x[1] | x |
| Slit-lamp microscopy | x | x | x | x | x | x | x | x | x[1] | x[1] | x[1] | x | x | x | x[1] | x[1] | x[1] | x[1] | x[1] | x[1] | x |
| ETDRS DRSS | | x | | | x | | | | | | | | x | | | | | | | | x |
| NEI-VFQ 25 | | x | | | x | | | | | | | | x | | | | | | | | x |
| WPAI | | x | | | x | | | | | | | | x | | | | | | | | x |
| Vital signs | x | x | x | x | x | x | x | x | x[1] | x[1] | x[1] | x | x | x | x[1] | x[1] | x[1] | x[1] | x[1] | x[1] | x |
| Adverse events | | x | x | x | x | x | x | x | x[1] | x[1] | x[1] | x | x | x | x[1] | x[1] | x[1] | x[1] | x[1] | x[1] | x |

**Fig 1. SWAN trial: Schedule of enrollment, interventions, and assessments.** *FA at day 1 only if not completed at screening; †To be completed at week 52 or later, at the visit next to a visit involving faricimab administration; ‡Indocyanine green angiography will only be performed if the investigator deems this necessary; x[1] Performed when visiting the site. DME, diabetic macular edema; ETDRS DRSS, Early Treatment Diabetic Retinopathy Study Diabetic Retinopathy Severity Scale; FA, fluorescein angiography; NEI VFQ-25, The 25-item National Eye Institute Visual Function Questionnaire; IOP, intraocular pressure; SD-OCT, spectral-domain optical coherence tomography; WPAI, Work Productivity and Activity Impairment Questionnaire.

defined as CST >325 μm (Spectralis SD-OCT; or >315 μm with Cirrus or Topcon SD-OCT) and pathologically relevant (defined as a cause of vision loss or other aggravation of the disease) intraretinal fluid (IRF) or subretinal fluid (SRF) in the study eye.

At week 12 and visits thereafter, patients with no active DME will follow an 8-week treatment extension regimen. Dosing intervals will be extended in 8-week increments up to a maximum of every 24 weeks (Q24W). Monitoring visits will occur every 4 weeks until week 24 and every 8 weeks thereafter. Faricimab will be administered on five subsequent occasions (at weeks 16, 32, 56, 80, and 104) unless active DME is observed.

Patients deemed to have active disease at week 12 will receive a dose of faricimab within a day and will then follow a 4-week treatment extension regimen. Patients deemed to have active disease during subsequent visits after week 12 will receive a dose of faricimab within a day and their previous dosing interval will be shortened by 4 weeks to a minimum of Q8W. Unless active disease is observed, the following and subsequent dosing intervals will be extended in 4-week increments up to a maximum of Q24W. Monitoring visits will occur every 4 weeks up

**Table 1. SWAN trial: Key ocular eligibility criteria for the study eye.**

| Ocular inclusion criteria | Ocular exclusion criteria |
|---|---|
| • Macular thickening secondary to DME involving the center of the fovea with CST ≥325 μm, as measured on Spectralis SD-OCT, or ≥315 μm, as measured on Cirrus SD-OCT or Topcon SD-OCT (or other equivalent OCTs) at screening<br><br>• BCVA of 0.0625 ~ 0.7 (decimal visual acuity) on visual acuity test conducted at screening<br><br>• Sufficiently clear optic media and adequate pupillary dilatation to allow acquisition of good quality CFP and other imaging modalities | • High-risk PDR in the study eye (using any of the following established criteria for high-risk PDR):<br> • Any vitreous or preretinal hemorrhageNeovascularization elsewhere ≥1/2 disc area within an area equivalent to the mydriatic ETDRS 7 fields on clinical examination or CFP<br> • Neovascularization at disc ≥1/3 disc area on clinical examination<br>• Tractional retinal detachment, preretinal fibrosis, vitreomacular traction syndrome, or epiretinal membrane involving the fovea or disrupting the macular architecture in the study eye<br>• Active rubeosis<br>• Uncontrolled glaucoma<br>• History of retinal detachment or macular hole (Stage 3 or 4)<br>• Aphakia or implantation of anterior chamber intraocular lens<br>• Intravitreal administration of anti-VEGF agents within 3 months before day 1 (applicable to patients whose study eyes were previously treated with intravitreal anti-VEGF agents), or any intravitreal administration of anti-VEGF agents to study eye before day 1 (applicable for treatment-naïve patients). Enrollment of patients who have a history of intravitreal administration of anti-VEGF agents should be no more than 25% of the total<br>• History of PRP, macular laser (focal, grid, or micropulse), any cataract surgery or treatment for complications of cataract surgery with steroids or YAG laser capsulotomy within 3 months before day 1<br>• Any other intraocular surgery (e.g., corneal transplantation, glaucoma filtration, pars plana vitrectomy, corneal transplant, or radiotherapy)<br>• Any intravitreal or periocular (subtenon) corticosteroid treatment within 6 months before day 1<br>• Treatment for other retinal diseases that can lead to macular edema<br>• Prior intravitreal administration of faricimab |

BCVA, best-corrected visual acuity; CFP, color fundus photography; CST, central subfield thickness; DME, diabetic macular edema;

ETDRS, Early Treatment Diabetic Retinopathy Study; PDR, proliferative diabetic retinopathy; PRP, panretinal photocoagulation;

SD-OCT, spectral-domain optical coherence tomography; VEGF, vascular endothelial growth factor; YAG, yttrium-aluminum-garnet.

to week 24 and every 8 weeks thereafter, depending on the dosing interval. The number of faricimab doses administered will depend on disease activity.

Outcome assessments will be completed at weeks 52, 56, 60, and 112.

## Visual assessments

Visual acuity tests will be completed before eye dilation for ophthalmic assessments. Corrected visual acuity will be assessed at 5 meters using Landolt rings. BCVA will be expressed in logMAR using the formula logMAR = log(1/d), where d = decimal visual acuity. If decimal visual acuity is <0.02, visual acuity will also be evaluated by finger counting, hand movement, or light perception. Visual acuity will be assessed during each hospital visit, which will be every 4 weeks up to week 24 and every 8 weeks thereafter. Assessments will also be completed at weeks 52, 56, and 60 for the primary endpoint analysis. Finger-counting tests will be performed within 15 minutes of faricimab administration to confirm the patient is not experiencing any immediate visual impairment.

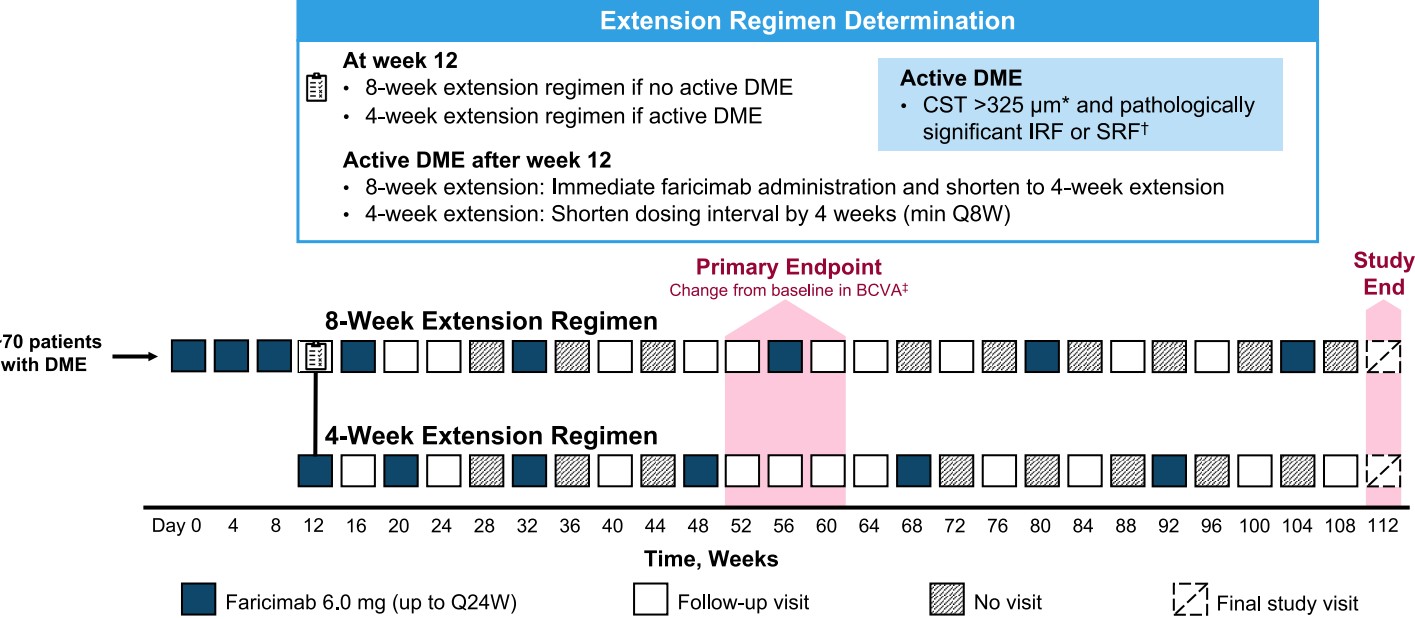

**Fig 2. Trial design overview.** * Spectralis SD-OCT >325 μm or Cirrus SD-OCT or Topcon SD-OCT >315 μm. † Considered clinically significant if deemed to be a cause of vision loss or other aggravation of the disease. ‡ Averaged over weeks 52, 56, and 60. BCVA, best-corrected visual acuity; CST, central subfield thickness; DME, diabetic macular edema; IRF, intraretinal fluid; Q8W, every 8 weeks; Q24W, every 24 weeks; SD-OCT, spectral-domain optical coherence tomography; SRF, subretinal fluid.

## Anatomic assessments

Key anatomic assessments will be completed during each hospital visit, which will be every 4 weeks up to week 24 and every 8 weeks thereafter. Assessments will also be made at the primary endpoint visits (weeks 52, 56, and 60).

Retinal anatomic features will be evaluated regularly using CFP and SD-OCT. Fluorescein angiography (FA) and OCT-angiography will be performed at baseline, week 12, the primary endpoint visits, and week 112. The investigator will measure retinal thickness within the center 1 mm diameter of the nine quadrants of the Early Treatment Diabetic Retinopathy Study (ETDRS) map as CST and check for the presence of IRF/SRF. Further imaging analyses, including the assessment of ETDRS Diabetic Retinopathy Severity Score (DRSS) based on CFP and FA, will be performed by the imaging clinical research organization (Micron Inc., Tokyo, Japan).

## Patient-reported outcomes

Patients will use smartphones or tablets for recording electronic patient-reported outcomes. Health-related quality of life (HRQoL) related to vision and work productivity will be assessed with the National Eye Institute Visual Function Questionnaire (NEI VFQ)-25 [11] and Work Productivity and Activity Impairment (WPAI) tools [12], respectively, at baseline, and weeks 12, 56, and 112. Patients will also be asked to record their subjective symptoms and any associated effect throughout the trial.

## Safety outcomes

All adverse events (AEs), ocular and nonocular, which occur between the first dose of faricimab and the last trial visit (or withdrawal of consent or loss to follow-up) will be reported,

regardless of their causality. The trial investigator will record AEs in the electronic case report form and the patient's medical record. AEs recorded at trial sites will also be reported back to the trial sponsor. Slit-lamp microscopy and indirect ophthalmoscopy will be completed after eye dilation, with abnormal findings recorded as complications or AEs.

## Trial outcomes

The SWAN trial key endpoints are summarized in Table 2. The primary endpoint is the change in BCVA from baseline at 1 year. The average change in BCVA over three timepoints (weeks 52, 56, and 60) will be evaluated to improve sensitivity and specificity because of the potential for intervisit variability [13, 14].

Change in BCVA over time and the proportions of patients who meet predefined thresholds for BCVA (i.e., decimal visual acuity $\geq 0.5$, $\geq 0.7$, $\geq 1.0$, or $\leq 0.1$) will be assessed as key secondary endpoints. Other key secondary endpoints will examine anatomic changes in the study eye, including change in CST from baseline over time, the proportion of patients with an absence of DME, IRF, and/or SRF over time, and the proportion of patients with a $\geq$two-step improvement in ETDRS-DRSS from baseline over time. To assess the durability of faricimab, the proportion of patients who achieve different faricimab treatment intervals will be monitored, as will the total number of faricimab doses.

The incidence and severity of ocular and nonocular AEs will be monitored throughout the trial.

Exploratory endpoints include change from baseline in macular leakage, number of microaneurysms in the macula, disorganization of the retinal inner layers, hyperreflective foci, and ellipsoid zone disruption over time, change from baseline in the subscales of NEI VFQ-25 over time, and the relationship between BCVA and CST change and HRQoL using the WPAI tool.

All SWAN trial prespecified endpoints are listed in S2 Table (see Supporting information).

**Table 2. SWAN trial: Key trial endpoints.**

| Primary endpoint | Secondary endpoints | Safety endpoints |
|---|---|---|
| • BCVA change (decimal visual acuity) from baseline at 1 year (averaged over weeks 52, 56, and 60) | • BCVA change from baseline over time<br>• CST change from baseline over time<br>• Proportion of patients meeting BCVA thresholds[*] over time<br>• Proportion of patients with absence of DME over time[†]<br>• Proportion of patients with absence of IRF and/or SRF over time<br>• Proportion of patients with $\geq$2-step improvement in DRSS from baseline over time<br>• Proportion of patients by faricimab treatment intervals | • Incidence and severity of ocular and nonocular AEs |

[*]Proportion of patients with/without $\geq$logMAR 0.3 BCVA improvement from baseline; proportion of patients with BCVA (decimal visual acuity) $\geq 0.5$, $\geq 0.7$, $\geq 1.0$, or $\leq 0.1$.

[†]Defined as CST $<325$ µm.

AE, adverse event; BCVA, best-corrected visual acuity; CST, central subfield thickness; DME, diabetic macular edema; DRSS, Diabetic Retinopathy Severity Score; IRF, intraretinal fluid; SRF, subretinal fluid.

## Statistical approaches

The trial aims to enroll a total of 70 patients with DME. Assuming that the true value is 1.48 (logMAR) and standard deviation is 1.48 (logMAR), based on data from the phase 3 YOSEMITE and RHINE trials, it is estimated that 45 patients will provide ≥80% probability that the point estimate of the primary endpoint is not more than 0.04 (logMAR) lower than the true value. Assuming the proportion of patients who achieve a dosing interval of at least Q16W is 50%, based on data from the phase 3 YOSEMITE and RHINE trials, 23 patients would provide ≥80% probability that the point estimate of the primary endpoint is not more than 0.04 (logMAR) lower than the true value in that population, with a 95% confidence interval of ± 4.48 letters. This is equivalent to less than one ETDRS line, which is considered clinically significant. A dropout rate of 35% is estimated for the first year of the trial, because of the burden of treatment costs and visits.

Data will be analyzed using SAS 9.4 (SAS Institute Inc., Cary, NC). BCVA analyses will follow the mixed effect model for repeated measures, which will include visit (categorical variable) and baseline BCVA (continuous variable) as fixed effects and will assume an unstructured covariance structure for modeling intrasubject error. For missing data, missing-at-random will be assumed.

## Trial status

The SWAN trial is currently recruiting patients. The first patient was enrolled August 2023.

## Discussion

The SWAN trial aims to evaluate the effect of real-world clinical practice T&E dosing of faricimab (up to Q24W) on efficacy, durability, and safety outcomes in patients with DME. The trial findings will provide information that may help optimize individualized treatment dosing in clinical practice and reduce the burden of treatment.

Different faricimab dosing regimens have been investigated previously in the phase 3 YOSEMITE and RHINE clinical trials in patients with DME [5]. Faricimab 6.0 mg Q8W after six initial Q4W doses and faricimab 6.0 mg personalized T&E up to Q16W dosing intervals (depending on DME activity) after ≥four initial Q4W doses were compared with aflibercept 2.0 mg Q8W after five initial Q4W doses. Visual gains with faricimab Q8W and T&E through 2 years were comparable with aflibercept Q8W. This was achieved despite fewer injections in the faricimab T&E up to Q16W treatment arm compared with aflibercept. More patients achieved absence of DME (CST <325 μm) and IRF with faricimab Q8W or T&E versus aflibercept. Over 60% of patients in the faricimab T&E arm achieved faricimab Q16W dosing at year 2. A *post hoc* analysis showed that most patients treated with faricimab T&E would also qualify for potential further extension of the interval to Q20W [9]. Collectively, these data suggest that a less frequent faricimab dosing regimen may maintain efficacy but with reduced burden on patients, caregivers, and healthcare providers.

The pathology of DME has been shown to improve with faricimab. For example, the number of existing microaneurysms and the formation of new microaneurysms have been shown to be reduced after three monthly faricimab injections in patients with DME [15]. Macular leakage area reduction and resolution of macular leakage have been shown to be greater with faricimab compared with aflibercept during the head-to-head treatment phase of YOSEMITE and RHINE [16]. Furthermore, in a *post hoc* analysis of YOSEMITE and RHINE, the median time to first DME <280 um was 16 weeks faster and with fewer injections with faricimab versus aflibercept [17]. Finally, in another *post hoc* analysis of YOSEMITE and RHINE, retinal hyperreflective foci volume and count reductions were found to be greater with faricimab

compared with aflibercept up to 48 weeks [18]. These findings suggest that dual Ang-2/VEGF-A inhibition with faricimab may improve blood vessel function and stabilize DME. To further evaluate this hypothesis, the presence of microaneurysms, macular leakage, IRF, and the presence of hyperreflective foci will be assessed in the SWAN trial.

This SWAN trial will explore the characteristics of patients for whom the faricimab dosing interval can be safely extended. The trial design permits a shortened initial monthly dosing phase with the fourth dose dependent on disease activity at week 12, with the potential for subsequent dosing intervals extended up to Q24W in the absence of active DME. CST >325 μm will be used as a marker of DME activity as preclinical data suggest that dual Ang-2/VEGF-A inhibition may promote vascular integrity and reduce retinal edema compared with VEGF inhibition alone [19]. *Post hoc* pooled data from the head-to-head YOSEMITE and RHINE dosing periods provided clinical evidence of this activity, with more patients treated with faricimab achieving resolution of macular leakage by week 16 than those treated with aflibercept [16]. For patients who qualify for maximal interval extension in the SWAN trial, the faricimab dosing regimen will be less intensive than that mandated by the current label [20]. A requirement for fewer clinic visits may improve adherence to DME treatment regimens and ultimately improve outcomes for patients.

It is important to acknowledge potential challenges that may arise during this trial. Specifically, there is a financial burden on patients related to the cost of faricimab as patients will need to pay for treatment. Additionally, there is a time burden as patients are required to attend clinic for testing at least every 2 months. Collectively, the financial and time burden may limit the number of patients interested in participating in this trial and may also affect patient dropout rate.

The SWAN trial will evaluate the efficacy, durability, and safety of faricimab as part of a T&E regimen that is adapted to patterns seen in routine clinical practice. The findings from this trial are expected to provide evidence to support individualized faricimab dosing regimens, with the potential to reduce the burden of frequent treatment visits on patients, caregivers, and healthcare systems.

## Supporting information

**S1 Table. SWAN trial eligibility criteria.** BCVA, best-corrected visual acuity; CFP, color fundus photography; CST, central subfield thickness; DME, diabetic macular edema; IMP, investigational medicinal product; PDR, proliferative diabetic retinopathy; PRP, panretinal photocoagulation; SD-OCT, spectral-domain optical coherence tomography; VEGF, vascular endothelial growth factor; YAG, yttrium-aluminum-garnet. *Postmenarchal women who have not reached postmenopausal status (amenorrhea for at least 12 consecutive months with no cause other than menopause), and are not permanently infertile by surgery (removal of ovaries, fallopian tubes and/or uterus) or other causes as determined by the investigator or co-investigator (e.g., Müllerian duct dysplasia) considered women of childbearing potential. According to this provision, women with unilateral tubal ligation are considered women of childbearing potential. †Examples of contraceptive methods with annual failure rates of less than 1% include bilateral tubal ligation, male sterilization, hormonal contraceptives that inhibit ovulation, hormone-releasing intrauterine devices, and copper-added intrauterine devices. The reliability of sexual abstinence should be evaluated with respect to the duration of the clinical research and each patient's preferences and normal lifestyle. Cyclic abstinence (calendar, ovulation day, symptomatic temperature, postovulation, etc.), and external ejaculation are not adequate contraceptive methods. ‡Systemic anti-VEGF therapy; systemic drugs known to cause macular edema (fingolimod, tamoxifen); intravitreal administration of anti-VEGF

agents (other than faricimab) into study eye; intravitreal, periocular (subtenon), steroid implants, or chronic topical ocular corticosteroids into study eye; photodynamic therapy to study eye; micropulse, and focal or grid photocoagulation in the study eye; vitreous surgery or PRP in the study eye; kallidinogenase (for improvement of symptoms of circulatory disturbance of the retinal choroid); other experimental therapies (except those comprising vitamins and minerals).
(PDF)

**S2 Table. SWAN trial prespecified endpoints.** AEs, adverse events; BCVA, best-corrected visual acuity; CST, central subfield thickness; DME, diabetic macular edema; ETDRS DRSS, Early Treatment Diabetic Retinopathy Study Diabetic Retinopathy Severity Scale; FA, fluorescein angiography; IRF, intraretinal fluid; NEI VFQ-25, The 25-item National Eye Institute Visual Function Questionnaire; OCT-A, optical coherence tomography angiography; Q24W, every 24 weeks; SD-OCT, spectral-domain optical coherence tomography; SRF, subretinal fluid; WPAI, Work Productivity and Activity Impairment Questionnaire.
(PDF)

**S1 Fig. YOSEMITE and RHINE trial design.** BCVA, best-corrected visual acuity; Q4W, every 4 weeks; Q8W, every 8 weeks; T&E, treat-and-extend. *, primary efficacy outcome: change in BCVA from baseline at 1 year, averaged over weeks 48, 52, and 56 (primary endpoint visits).
(PDF)

**S1 Checklist. SPIRIT checklist.**
(DOC)

**S1 Protocol. SWAN protocol.**
(PDF)

## Author Contributions

**Conceptualization:** Takao Hirano, Toshinori Murata, Shintaro Nakao, Shigeo Yoshida.

**Investigation:** Takao Hirano, Toshinori Murata, Shintaro Nakao, Masahiko Shimura, Miho Nozaki, Kiyoshi Suzuma, Taiji Nagaoka, Masahiko Sugimoto, Yoshihiro Takamura, Tomoaki Murakami, Shigeo Yoshida.

**Methodology:** Takao Hirano, Toshinori Murata, Shintaro Nakao, Jun Tsujimura, Shigeo Yoshida.

**Project administration:** Jun Tsujimura.

**Resources:** Jun Tsujimura.

**Supervision:** Takao Hirano, Toshinori Murata, Shintaro Nakao, Keisuke Iwasaki, Shigeo Yoshida.

**Writing – original draft:** Takao Hirano, Toshinori Murata, Shintaro Nakao, Masahiko Shimura, Miho Nozaki, Kiyoshi Suzuma, Taiji Nagaoka, Masahiko Sugimoto, Yoshihiro Takamura, Tomoaki Murakami, Keisuke Iwasaki, Jun Tsujimura, Shigeo Yoshida.

**Writing – review & editing:** Takao Hirano, Toshinori Murata, Shintaro Nakao, Masahiko Shimura, Miho Nozaki, Kiyoshi Suzuma, Taiji Nagaoka, Masahiko Sugimoto, Yoshihiro Takamura, Tomoaki Murakami, Keisuke Iwasaki, Jun Tsujimura, Shigeo Yoshida.

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
