## [Decision Letter · Decision Letter 0]

15 Jul 2024

PONE-D-24-20969Optimization of individualized faricimab dosing for patients with diabetic macular edema: protocol for the SWAN open-label, single-arm clinical trialPLOS ONE

Dear Dr. Hirano,

Thank you for submitting your manuscript to PLOS ONE. After careful consideration, we feel that it has merit but does not fully meet PLOS ONE’s publication criteria as it currently stands. Therefore, we invite you to submit a revised version of the manuscript that addresses the points raised during the review process.

We look forward to receiving your revised manuscript.

Kind regards,

Jiro Kogo

Academic Editor

PLOS ONE

Journal Requirements:

3. Thank you for stating the following financial disclosure: "Supported by Chugai Pharmaceutical Co., Ltd. The sponsor participated in the study design; the writing of the report; and the decision to submit the paper for publication. Funding was provided by Chugai Pharmaceutical Co., Ltd for the study and third-party writing assistance, which was supplied by Trishan Gajanand, PhD, and Nicole Tom, PhD, of Envision Pharma Group." 

4. Thank you for stating the following in the Competing Interests section: "Takao Hirano: Speaker Fees: Bayer, Canon, Chugai Pharmaceutical Co., Ltd., Kowa, Novartis Pharma KK, Santen Pharmaceutical Co., Senju, ZEISS

Toshinori Murata: Speaker Fees: Bayer, Novartis, Santen, Zeiss Meditec: Consultant: Boehringer Ingelheim, Chugai Pharmaceutical Co., Ltd., Hoya, Kowa, Wakamoto

Shintaro Nakao: Consultant: Alcon, Boehringer Ingelheim, Novartis, Riverfield; Speaker Fees: Bayer, Boehringer Ingelheim, Chugai Pharmaceutical Co., Ltd., Hoya, Kowa, Machida, Mitsubishi Tanabe, Novartis, Novo Nordisk, Otsuka, Santen, Senju, Wakamoto

Masahiko Shimura: Consultant: Bayer, Boehringer Ingelheim, Chugai, HOYA, Nikki HD, Roche, Wakamoto; Lecture Fees: Bayer, Chugai, Kowa, Novartis, Otsuka, Senju

Miho Nozaki: Speaker Fees: Bayer, Canon, Chugai Pharmaceutical Co., Ltd., Kowa, Nikon, Novartis Pharma KK, Santen, Senju, Sumitomo Pharma Co., Ltd., Topcon Medical, Wakamoto

Kiyoshi Suzuma: Speaker Fees: AMO, Alcon, Bayer, Chugai Pharmaceutical Co., Ltd., HOYA, Kowa, Novartis, Senju

Consultant: Senju, Chugai Pharmaceutical Co., Ltd., Boehringer Ingelheim

Taiji Nagaoka: Consultant: Boehringer Ingelheim, Novartis, TES Holdings; Speaker Fees: Bayer, Chugai Pharmaceutical Co., Ltd., Hoya, Kowa, Mitsubishi Tanabe, Novartis, Santen, Senju, Wakamoto; Research Fees: Daicel, Kowa, LTT, Santen

Masahiko Sugimoto: Research Fees: Alcon Japan, Bayer, Chugai Pharmaceutical Co., Ltd., Novartis; Speaker Fees: Alcon Japan, Bayer, Kowa, Novartis, Senju, Wakamoto

Yoshihiro Takamura: Lecture Fees: Chugai Pharmaceutical Co., Ltd. 

Tomoaki Murakami: Speaker Fees: Bayer, Canon, Chugai Pharmaceutical Co., Ltd., Johnson & Johnson, Kowa, Novartis Pharma KK, Santen; Consultant: Boehringer Ingelheim

Keisuke Iwasaki: Employee: Chugai Pharmaceutical Co., Ltd. 

Jun Tsujimura: Employee: Chugai Pharmaceutical Co., Ltd.

Shigeo Yoshida: Consultant: Chugai Pharmaceutical Co., Ltd., Novartis; Speaker Fees: Bayer, Chugai Pharmaceutical Co., Ltd., Novartis, Senju; Research Fees: Kowa, Otsuka, Senju 

"

5. Please provide a complete Data Availability Statement in the submission form, ensuring you include all necessary access information or a reason for why you are unable to make your data freely accessible. If your research concerns only data provided within your submission, please write "All data are in the manuscript and/or supporting information files" as your Data Availability Statement.

Reviewers' comments:

Reviewer's Responses to Questions

**Comments to the Author**

1. Does the manuscript provide a valid rationale for the proposed study, with clearly identified and justified research questions?

Reviewer #1: Yes

Reviewer #2: Yes

2. Is the protocol technically sound and planned in a manner that will lead to a meaningful outcome and allow testing the stated hypotheses?

Reviewer #1: Yes

Reviewer #2: Partly

3. Is the methodology feasible and described in sufficient detail to allow the work to be replicable?

Reviewer #1: Yes

Reviewer #2: Yes

4. Have the authors described where all data underlying the findings will be made available when the study is complete?

Reviewer #1: Yes

Reviewer #2: No

5. Is the manuscript presented in an intelligible fashion and written in standard English?

Reviewer #1: Yes

Reviewer #2: Yes

6. Review Comments to the Author

You may also provide optional suggestions and comments to authors that they might find helpful in planning their study.

Reviewer #1: This manuscript is a introduction of SWAN trial, a 2-year, open-label, single-arm, interventional, multicenter trial enrolling adults with center-involving DME treated with faricimab 0.6mg.

Improvement of vision outcomes and reduction of treatment burden are warranted for anti-VEGF therapy and faricimab is expected to resolve such unmet needs.

The SWAN trial is excellently organized prospective study. I am looking forward to the authors’ continue success.

I think that the manuscript is suitable for the publication in the PLOS ONE.

Reviewer #2: From the statistical viewpoint, it is a curiosity that the authors did not consider possibility of using an adaptive design for a single arm trial. It is helpful such a discussion be included in the paper. The adaptive design has the potential of requiring fewer patients to be enrolled and less time to conclude whether the proposed treatment is effective or not. For example, a most common trial of this kind is the celebrated Simon Two-Stage design for a Phase 2 trials discussed and implemented in clinical trials with many variations thereof. It has two stages and assumes the primary endpoint is binary (responders or not) and depending on the observed proportion of responders in stage 1, the trial may or may not move to the second stage to continue to assess the efficacy of the treatment. Here a null hypothesized success rate is postulated along with an alternative hypothesis that postulates a higher rate of success. The strategy is that if the null hypothesis is not met, the treatment is not effective and the trial terminates. If it is not rejected the trial proceeds to the second stage. In each stage, the number of patients and number of responders have to be determined after the user-selected type 1 and 2 errors for testing the hypotheses are specified. The design problem is then to determine the number of patients and the number of responders required in Stage 1, and if the trial proceeds to Stage 2, the additional number of patients needed and the additional number of responders needed to conclude efficacy of the treatment.

Clarifications:

It was stated in the abstract and elsewhere that “The primary endpoint is change in best-corrected visual acuity (BCVA) from baseline at 1 year (averaged over weeks 52, 56, and 60)” This seems unclear to me what is meant by the averaging referred to in the parentheses. Please clarify.

In the abstract, not clear secondary/exploratory endpoints mean. Are you treating secondary and exploratory endpoints equivalent? If not, make clear what are secondary endpoints and what are exploratory endpoints. The result sections can use a rewrite to make clearer each of the variables involved and the threshold for success – the sentence right now is hard to understand because it is long and somewhat convoluted.

The description of the proposed trial and the two previous trials (YOSEMITE (NCT03622580) and RHINE (NCT03622593)) should be contrasted in a diagram or schematic form to facilitate understanding of the treatment regimen involved. Right now, they are wordy going from lines 69 to 90, which can cloud comprehension.

Statistical analyses:

In the sample size calculation, the study is powered for detecting a postulated difference. Can the author discussed whether the postulated difference of interest is a meaningful outcome from the clinical perspective since a significant difference may not correspond to a clinically meaningful difference.

A 35% dropout rate is anticipated in the first year. Please justify where this number is coming from and whether the proposed analyses account for possible values other than a 35% dropout rate.

There are several secondary variables to be evaluated, such as determining the proportions of patients who meet predefined thresholds for BCVA (i.e., decimal visual acuity ≥0.5, ≥0.7, ≥1.0, or ≤0.1) over time, in addition to many others. Is there going to be a single composite measure that describes whether the patients did well or not, based on the secondary variables?

Remaining comments pertain to the last paragraph prior to “Trial status” on page 25, where the authors stated that “BCVA analyses will follow the mixed effect model for repeated measures, which will include visit (categorical variable) and baseline BCVA (continuous variable) as fixed effects and will assume an unstructured covariance structure for modeling intrasubject error. For missing data, missing-at-random will be assumed.”

a. Why is the variable 'visit' is treated as a categorical variable and not as a continuous variable in real time? In the trial, patients are supposed to visit at designated time points, and they rarely do so in practice. What happens if patients show up at different time points? What is the threshold before the reading is considered invalid because the timing is unacceptably different from that specified in the protocol? For example, patients are required to visit a clinic for testing at least every 2 months – what happens if patient show up ½ month late? Is this a valid data point? Further, treating the variable visit as a categorical variable has implicit constraints imposed on the interpretation of the results.

b. Please make clearer what are the random components in the mixed effects model alluded to?

c. Are the authors going to fit other covariance structures and select the best fitting model for prediction purposes?

d. To ensure reproducibility, please mention what statistical package or software is going to be used to analyze the data.

e. Missing data are assumed to be missing-at-random. It is helpful to elaborate how the authors are going to check whether such an assumption is valid – in particular, are statistical diagnostic tools going to be employed to ascertain whether such an assumption seems valid, and if so how?

7. PLOS authors have the option to publish the peer review history of their article (what does this mean?). If published, this will include your full peer review and any attached files.

Reviewer #1: **Yes: **Taiichi Hikichi

Reviewer #2: No

---

## [Author Response · Author response to Decision Letter 0]

9 Sep 2024

Editor

Thank you for stating the following financial disclosure: "Supported by Chugai Pharmaceutical Co., Ltd. The sponsor participated in the study design; the writing of the report; and the decision to submit the paper for publication. Funding was provided by Chugai Pharmaceutical Co., Ltd for the study and third-party writing assistance, which was supplied by Trishan Gajanand, PhD, and Nicole Tom, PhD, of Envision Pharma Group." 

Author comments

The funder participated in the study design, data collection and analysis, the decision to publish, and in the preparation of the manuscript – this statement has been included in the cover letter as requested.

Changes to manuscript

No change to the manuscript – the role of the funder has been amended in the cover letter.

Editor

Thank you for stating the following in the Competing Interests section: "Takao Hirano: Speaker Fees: Bayer, Canon, Chugai Pharmaceutical Co., Ltd., Kowa, Novartis Pharma KK, Santen Pharmaceutical Co., Senju, ZEISS

Toshinori Murata: Speaker Fees: Bayer, Novartis, Santen, Zeiss Meditec: Consultant: Boehringer Ingelheim, Chugai Pharmaceutical Co., Ltd., Hoya, Kowa, Wakamoto

Shintaro Nakao: Consultant: Alcon, Boehringer Ingelheim, Novartis, Riverfield; Speaker Fees: Bayer, Boehringer Ingelheim, Chugai Pharmaceutical Co., Ltd., Hoya, Kowa, Machida, Mitsubishi Tanabe, Novartis, Novo Nordisk, Otsuka, Santen, Senju, Wakamoto

Masahiko Shimura: Consultant: Bayer, Boehringer Ingelheim, Chugai, HOYA, Nikki HD, Roche, Wakamoto; Lecture Fees: Bayer, Chugai, Kowa, Novartis, Otsuka, Senju

Miho Nozaki: Speaker Fees: Bayer, Canon, Chugai Pharmaceutical Co., Ltd., Kowa, Nikon, Novartis Pharma KK, Santen, Senju, Sumitomo Pharma Co., Ltd., Topcon Medical, Wakamoto

Kiyoshi Suzuma: Speaker Fees: AMO, Alcon, Bayer, Chugai Pharmaceutical Co., Ltd., HOYA, Kowa, Novartis, Senju

Consultant: Senju, Chugai Pharmaceutical Co., Ltd., Boehringer Ingelheim

Taiji Nagaoka: Consultant: Boehringer Ingelheim, Novartis, TES Holdings; Speaker Fees: Bayer, Chugai Pharmaceutical Co., Ltd., Hoya, Kowa, Mitsubishi Tanabe, Novartis, Santen, Senju, Wakamoto; Research Fees: Daicel, Kowa, LTT, Santen

Masahiko Sugimoto: Research Fees: Alcon Japan, Bayer, Chugai Pharmaceutical Co., Ltd., Novartis; Speaker Fees: Alcon Japan, Bayer, Kowa, Novartis, Senju, Wakamoto

Yoshihiro Takamura: Lecture Fees: Chugai Pharmaceutical Co., Ltd. 

Tomoaki Murakami: Speaker Fees: Bayer, Canon, Chugai Pharmaceutical Co., Ltd., Johnson & Johnson, Kowa, Novartis Pharma KK, Santen; Consultant: Boehringer Ingelheim

Keisuke Iwasaki: Employee: Chugai Pharmaceutical Co., Ltd. 

Jun Tsujimura: Employee: Chugai Pharmaceutical Co., Ltd.

Shigeo Yoshida: Consultant: Chugai Pharmaceutical Co., Ltd., Novartis; Speaker Fees: Bayer, Chugai Pharmaceutical Co., Ltd., Novartis, Senju; Research Fees: Kowa, Otsuka, Senju"

Author comments

We confirm that the author competing interests do not alter our adherence to PLOS ONE policies on sharing data and materials – this statement has been included in the cover letter as requested. In addition, we have added the following data sharing statement to the submission form. 

The datasets generated during and/or analysed during the current study are available from the corresponding author on reasonable request.

Editor

Please provide a complete Data Availability Statement in the submission form, ensuring you include all necessary access information or a reason for why you are unable to make your data freely accessible. If your research concerns only data provided within your submission, please write "All data are in the manuscript and/or supporting information files" as your Data Availability Statement.

Author comments

We have added the following data sharing statement to the submission form.

The datasets generated during and/or analysed during the current study are available from the corresponding author on reasonable request.

Editor

Please include captions for your Supporting Information files at the end of your manuscript, and update any in-text citations to match accordingly. Please see our Supporting Information guidelines for more information: http://journals.plos.org/plosone/s/supporting-information.

Author comments

Captions for the supporting information files have now been included at the end of the manuscript.

Changes to manuscript

The following has been added to the end of the manuscript:

“Supporting Information

Table S1. SWAN trial eligibility criteria

Table S2. SWAN trial prespecified endpoints

Figure S1: YOSEMITE and RHINE Trial Design”

Editor

Author comments

We have updated reference 6 (Wong TY et al. Ophthalmology. 2024) as this was previously online only but has now been published.

We have updated reference 8 (Shimura M et al. Jpn J Ophthalmol. 2024) as this was previously under review but is now published online.

Changes to manuscript

References 6 and 8 have been updated as follows:

6. Wong TY, Haskova Z, Asik K, Baumal CR, Csaky KG, Eter N, et al. Faricimab Treat-and-Extend for Diabetic Macular Edema: Two-Year Results from the Randomized Phase 3 YOSEMITE and RHINE Trials. Ophthalmology. 2024;131(6):708-23. Epub 2024/01/02. doi: 10.1016/j.ophtha.2023.12.026.

8. Shimura M, Oh H, Ueda T, Kitano S, Mitamura Y, Sato J, et al. Efficacy, durability, and safety of faricimab with extended dosing up to every 16 weeks in diabetic macular edema: 2-year results from the Japan subgroup of the phase 3 YOSEMITE trial. Jpn J Ophthalmol. 2024. Epub 2024/07/31. doi: 10.1007/s10384-024-01078-y.

Reviewer 1

This manuscript is a introduction of SWAN trial, a 2-year, open-label, single-arm, interventional, multicenter trial enrolling adults with center-involving DME treated with faricimab 0.6mg.

Improvement of vision outcomes and reduction of treatment burden are warranted for anti-VEGF therapy and faricimab is expected to resolve such unmet needs.

The SWAN trial is excellently organized prospective study. I am looking forward to the authors’ continue success.

I think that the manuscript is suitable for the publication in the PLOS ONE.

Author comments

We thank the Reviewer for these comments.

Reviewer 2

From the statistical viewpoint, it is a curiosity that the authors did not consider possibility of using an adaptive design for a single arm trial. It is helpful such a discussion be included in the paper. The adaptive design has the potential of requiring fewer patients to be enrolled and less time to conclude whether the proposed treatment is effective or not. For example, a most common trial of this kind is the celebrated Simon Two-Stage design for a Phase 2 trials discussed and implemented in clinical trials with many variations thereof. It has two stages and assumes the primary endpoint is binary (responders or not) and depending on the observed proportion of responders in stage 1, the trial may or may not move to the second stage to continue to assess the efficacy of the treatment. Here a null hypothesized success rate is postulated along with an alternative hypothesis that postulates a higher rate of success. The strategy is that if the null hypothesis is not met, the treatment is not effective and the trial terminates. If it is not rejected the trial proceeds to the second stage. In each stage, the number of patients and number of responders have to be determined after the user-selected type 1 and 2 errors for testing the hypotheses are specified. The design problem is then to determine the number of patients and the number of responders required in Stage 1, and if the trial proceeds to Stage 2, the additional number of patients needed and the additional number of responders needed to conclude efficacy of the treatment.

Author comments

We thank the reviewer for these comments and for this suggestion. Unfortunately, we will not be able to change the study design because the SWAN trial started in August 2023.

We did not use an adaptive study design for this study as the purpose of the trial is to evaluate the efficacy and durability of faricimab during the maintenance period under real-world clinical conditions for DME. Specifically, accumulating long-term data from all cases itself is of importance in this study, and we are not looking to introduce criteria for discontinuation due to inefficacy. In addition, we aim to assess the impact of adjusting the dosing interval on efficacy. If we included discontinuation criteria for inefficacy, there is a possibility that these aspects may not be adequately evaluated. For these reasons, the study was designed as described.

Reviewer 2

It was stated in the abstract and elsewhere that “The primary endpoint is change in best-corrected visual acuity (BCVA) from baseline at 1 year (averaged over weeks 52, 56, and 60)” This seems unclear to me what is meant by the averaging referred to in the parentheses. Please clarify.

Author comments

The visit when the study drug is administered varies by patient, with some patients receiving faricimab at week 48 and others at week 52. Therefore, to minimize the effect of differences in time from last treatment, and to account for best-corrected visual acuity (BCVA) variability over time, the primary endpoint was averaged over weeks 52, 56, and 60. This approach was also used in the phase 3 YOSEMITE and RHINE trials (Wykoff CC et al. Lancet. 2022. 399(10326):741-755. doi: 10.1016/S0140-6736(22)00018-6.)

Reviewer 2

In the abstract, not clear secondary/exploratory endpoints mean. Are you treating secondary and exploratory endpoints equivalent? If not, make clear what are secondary endpoints and what are exploratory endpoints. The result sections can use a rewrite to make clearer each of the variables involved and the threshold for success – the sentence right now is hard to understand because it is long and somewhat convoluted.

Author comments

We have removed the term “exploratory” from the Results section of the Abstract for clarity. In addition, the text “proportion of patients who achieve Q24W dosing” has been removed.

Changes to manuscript

The second sentence of the Results section of the Abstract has been adjusted to the following:

“Key secondary endpoints include: change from baseline in BCVA, CST, and National Eye Institute Visual Function Questionnaire scores over time; proportion of patients with BCVA (decimal visual acuity) ≥0.5, ≥0.7, ≥1.0, or ≤0.1; proportion of patients with absence of DME, and IRF and/or SRF over time.”

Reviewer 2

The description of the proposed trial and the two previous trials (YOSEMITE (NCT03622580) and RHINE (NCT03622593)) should be contrasted in a diagram or schematic form to facilitate understanding of the treatment regimen involved. Right now, they are wordy going from lines 69 to 90, which can cloud comprehension.

Author comments

The study design schematic for the YOSEMITE and RHINE trials has been added to the manuscript as a supplementary figure. 

We have also edited the text describing the SWAN and YOSEMITE/RHINE trials in the Introduction for clarity. 

Changes to manuscript

We have removed the following from the second paragraph of the Introduction:

“In addition, more faricimab-treated patients than aflibercept-treated patients achieved intraretinal fluid (IRF) absence and central subfield thickness (CST) <325 µm through year 2 [6].”

We have adjusted the final sentence of the second paragraph of the Introduction to the following:

“Furthermore, a subsequent YOSEMITE and RHINE post hoc analysis found that 56% of patients who achieved Q16W faricimab dosing met the criteria for potential extension to every 20 weeks (Q20W) dosing [9].”

We have added the YOSEMITE/RHINE trial design figure as supplementary with the following title:

“Figure S1: YOSEMITE and RHINE Trial Design”

Reviewer 2

Statistical analyses:

In the sample size calculation, the study is powered for detecting a postulated difference. Can the author discussed whether the postulated difference of interest is a meaningful outcome from the clinical perspective since a significant difference may not correspond to a clinically meaningful difference.

Author comments

A clinically meaningful difference is 5 letters on Early Treatment Diabetic Retinopathy Study, which is an improvement of approximately 0.1 when converted to logMAR. The SWAN study was designed to assume an improvement of 0.22 when converted to logMAR, and the sample size was determined with consideration for this clinically meaningful difference.

Reviewer 2

A 35% dropout rate is anticipated in the first year. Please justify where this number is coming from and whether the proposed analyses account for possible values other than a 35% dropout rate.

Author comments

A dropout rate of 35% is estimated for the first year of the SWAN trial due to the burden of treatment costs and visits.

In the YOSEMITE/RHINE trials, the dropout rate over one year was approximately 10%; however, in the current study, since patients will be responsible for the cost of faricimab, and because of the burden of hospital visits, a dropout rate of 35% is anticipated during the first year. The target number of patients was set at 70.

Reviewer 2

There are several secondary variables to be evaluated, such as determining the proportions of patients who meet predefined thresholds for BCVA (i.e., decimal visual acuity ≥0.5, ≥0.7, ≥1.0, or ≤0.1) over time, in addition to many others. Is there going to be a single composite measure that describes whether the patients did well or not, based on the secondary variables?

Author comments

No single composite measure based on the secondary variables will be used in this study. Each binary endpoint was designed to evaluate the improvement and maintenance of visual acuity using multiple thresholds for BCVA.

Reviewer 2

Remaining comments pertain to the last paragraph prior to “Trial status” on page 25, where the authors stated that “BCVA analyses will follow the mixed effect model for repeated measures, which will include visit (categorical variable) and baseline BCVA (continuous variable) as fixed effects and will assume an unstructured covariance structure for modeling intrasubject error. For missing data, missing-at-random will be assumed.”a. Why is the variable 'visit' is treated as a categorical variable and not as a continuous variable in real time? In the trial, patients are supposed to visit at designated time points, and they rarely do so in practice. What happens if patients show up at different time points? What is the threshold before the reading is considered invalid because the timing is unacceptably different 

---

## [Editor Report · Decision Letter 1]

20 Sep 2024

Optimization of individualized faricimab dosing for patients with diabetic macular edema: protocol for the SWAN open-label, single-arm clinical trial

PONE-D-24-20969R1

Dear Dr. Hirano

We’re pleased to inform you that your manuscript has been judged scientifically suitable for publication and will be formally accepted for publication once it meets all outstanding technical requirements.

Kind regards,

Jiro Kogo

Academic Editor

PLOS ONE

---

## [Editor Report · Acceptance letter]

1 Oct 2024

PONE-D-24-20969R1 

PLOS ONE

Dear Dr. Hirano, 

I'm pleased to inform you that your manuscript has been deemed suitable for publication in PLOS ONE. Congratulations! Your manuscript is now being handed over to our production team.

Kind regards, 

on behalf of

Prof. Jiro Kogo 

Academic Editor

PLOS ONE